# A Microsatellite Genotyping-Based Genetic Study of Interspecific Hybridization between the Red and Sika Deer in the Western Czech Republic

**DOI:** 10.3390/ani11061701

**Published:** 2021-06-07

**Authors:** Lenka Štohlová Putnová, Radek Štohl, Martin Ernst, Kateřina Svobodová

**Affiliations:** 1Department of Animal Morphology, Physiology and Genetics, Faculty of AgriScience, Mendel University in Brno, Zemědělská 1, 613 00 Brno, Czech Republic; lenka.putnova@mendelu.cz; 2Department of Control and Instrumentation, Faculty of Electrical Engineering and Communication, Brno University of Technology, Technická 12, 616 00 Brno, Czech Republic; 3Department of Forest Protection and Wildlife Management, Faculty of Forestry and Wood Technology, Mendel University in Brno, Zemědělská 3, 613 00 Brno, Czech Republic; martin.ernst@mendelu.cz (M.E.); katerina.svobodova@mendelu.cz (K.S.)

**Keywords:** genetic structure, introgression, microsatellite variability, hybridization, sika-red deer hybrid

## Abstract

**Simple Summary:**

The sika deer is a very flexible invasive species, capable of living dynamically in both large forests and mixed environment characterized by a prevalence of agricultural land. The Japanese sika deer was introduced to the Czech Republic at the end of the 19th century. The success of an introduced alien species may consist in their hybridizing with closely related taxa. Where few barriers to gene flow exist, rapid introgression of genetic traits from one species into another frequently occurs. The current Czech sika populations embody the most abundant and expanding group in continental Europe. In western Bohemia, we confirmed the interspecific hybridization with the native red deer. In this context, the red deer gene pool is endangered. The animals proliferate steadily in all directions and will most probably spread all over the Czech Republic if no major, timely changes in game management are adopted.

**Abstract:**

Although inter-species hybrids between the red and sika deer can be phenotypically determined only exceptionally, there is the eventuality of identification via molecular genetic analysis. We used bi-parentally inherited microsatellite markers and a Bayesian statistical framework to re-examine the proportion of hybrids in the Czech red and sika deer populations. In total, 123 samples were collected, and the nuclear dataset consisted of 2668 allelic values. The number of alleles per locus ranged from 10 (*BM1818*) to 22 (*BM888* and *T193*), yielding the mean of 16 alleles per locus across the deer. The mean allelic diversity of the red deer markedly exceeded that of the Japanese sika deer. Interspecific hybrids were detected, enabling us to confirm the genetic introgression of the sika deer into the red deer populations and vice versa in western Bohemia. The mean hybrid score equaled 10.6%, with 14.3% of the hybrids being among red deer–like individuals and 6.7% among sika-like ones. At two western Bohemian locations, namely, Doupovské hory and Slavkovský les, the total percentages of hybrid animals equaled 18.8 and 8.9, respectively. No red deer alleles were detected in the sika populations of the subregions of Kladská, Žlutice, and Lány. The NeighborNet network clearly separated the seven red and sika deer sampling populations according to the geography. The knowledge gained from the evaluated data is applicable in hunting management to reduce hybridization with the European deer.

## 1. Introduction

Hybridization is a common phenomenon in plants, birds, fish, and many other taxa [1,2,3]. When, however, hybridization threatens native species, it becomes important to understand the evolutionary mechanisms and develop sensible management and conservation measures [4,5,6].

The human translocation (or illegal introduction) of an exotic non-native species into an ecosystem [7] often results in anthropogenically induced hybridization between the introduced species and the related native genera [8,9,10]. In this context, we focused on the red deer (*Cervus elaphus* Linnaeus 1758), which is native to Europe, and the sika deer (*Cervus nippon* Temminck 1838), an originally eastern Asian species introduced to diverse parts of the world outside its native range [11,12]. While red deer and sikas differ in body size, a variety of other phenotypic traits, and chromosome number (red 2*N =* 68; sika 2*N =* 64, [13]), the hybrids are fertile, and, therefore, can backcross with their parental species both in captivity and in the wild [8,12,14,15,16,17]. The offspring of such multiple crosses are very difficult to identify according to the morphological traits; essentially, their morphological differentiation from the standard sika/red deer is unfeasible [12,18,19]. However, modern molecular genetic techniques help us to detect the proportion of hybrid individuals [15,16,17,20,21,22,23,24,25,26]. At present, in the Czech Republic, molecular genetic analyses are nevertheless applied only scarcely within this topic, especially due to being costly and logistically problematic. This paper was therefore conceived to examine the presumed crossbreeding of sikas with native red deer by exploiting a genetic panel of microsatellite markers. Microsatellites are popular markers for the study of geographical structure and gene flow because they are co-dominant, multi-allelic, and abundant with a wide genome coverage [22,27]. To date, the hybridization of the deer in free-ranging areas has been explored mainly in the UK, where the morphometric and subsequent genetic analyses proved that the actual existence of the hybrids constitutes not a merely potential threat, but rather a real, unexpectedly widespread problem [15,16,20,23]. The presence of sika-red deer hybrids has also been confirmed in Poland [21], Germany [28], and Lithuania [24] via modern molecular methods. Importantly, pure species (native and introduced) may no longer exist: the evolutionary trajectory of a species may be radically changed through the introgression of large numbers of alleles, a process that effectively endangers that species’ genetic integrity [17].

The European red deer ranges among the largest representatives of its family: After the European elk (*Alces alces*), it is the largest wild ungulate in Bohemia and Moravia. Besides the roe deer (*Capreolus capreolus*), the European red deer embodies one of the original even-toed ungulate species in Bohemia and Moravia. In some areas, the density dropped to zero, with the population wiped out and subsequently recovering through incoming animals of foreign origin [29]. Considering the Czech Republic, the red deer, originally a forest steppe species, now live mainly in large forests at medium and high altitudes. A distribution map of the red deer can be found in a dedicated paper by Anděra [30]. While the number of individuals hunted down in the 1960s and 1970s reached approximately 8000, in 1988, the total count already exceeded 20,000, a figure second only to that of almost 27,000 in 1993 (these variations being due to political changes in the country). Then, there followed a drop back to around 20,000 individuals; in 2019/2020, the indicator equaled 29,017 [29,31].

The Japanese sika deer is considered an exotic invasive alien species, introduced to the Czech Republic only at the end of the 19th century; the first import took place in 1891 [32]. In the late nineteenth and early twentieth centuries, sikas were frequently delivered to populate deer parks [11,12]. Between the early 1920s and late 1940s, namely, during a period of major political turbulence in the country, some deer parks were destroyed, and the sikas escaped [33,34]. The species then spread farther, especially from the parks in the Bohemian region of Pilsen and the Moravian subregion of Bouzov. This process is reflected in the current distribution of sikas, which have become particularly prominent within western and northwestern Bohemia and, to a slightly lesser extent, northwestern Moravia. The constantly very high population size and density of the species increase annually, and the overall occupied area expands rapidly.

Evidence from the 1950s indicates that the largest population was hosted by Plzeňská pahorkatina (The Pilsen Uplands) [35,36,37] and that the effort to cull the animals to quickly attain the standardized game stock eventually prompted the sikas to spread beyond the area in a relatively rapid manner. The species then gradually invaded Český les (The Upper Palatinate Forest), Slavkovský les (Slavkov Forest), Doupovské hory (The Doupov Mountains) and, partially, Brdy (The Brdy Hills), Slavkovský les and Doupovské hory have included sika populations since the early and middle 1960s. In the latter region and its subareas, the species did not constitute a major concern initially, thus gaining dynamic potential for a future surge in the game stocks and transforming the region into a refuge sought after by surplus animals. Over two decades, the originally attractive game became a virtually unsolvable problem [35]. Although the necessity to adopt convenient measures without delay to prevent undesired territorial expansion and stock growth was emphasized by Wolf and Vavruněk [38] already almost half a century ago, the situation has clearly not improved: In sharp contrast, sikas have still moved farther to increase in number across western Bohemia, and the trend apparently continues [37,39].

Between 2003 and 2010, the territorial spread of sikas locally distributed across the Czech Republic involved, on average, a further 55 thousand hectares every year [40]. At present, this species is found over almost a third of the country. In 2016, compared to 2006, the territory had enlarged by 19%, and the culling rate had risen by an incredible 130%, from 6200 to 14,400 animals [41]. The current Czech sika populations embody the most abundant and expanding group in continental Europe [12,37,42]; such conditions are made possible mainly by ample food sources and shelter. In this context, however, it also has to be emphasized that a very significant factor consists in the applied game management approach: in terms of hunting, sikas are widely preferred, prevailing on many hunting grounds to improve the shooting opportunities and to intensify the related benefits. An updated, sika-related map illustrative of the current situation in the Czech Republic was presented by Anděra [43]. The regional count of the deer increased exponentially during the last several decades, with the real population numbers probably exceeding those estimated and reported in the yearly hunting statistics [39]. Currently, the Czech Republic’s sika population, based on hunters’ reports for 2019/2020 includes 17,535 individuals [31]. Sikas may generate considerable environmental impact due to their capability of causing significant damage in forestry, agriculture, and habitat structures [36,44,45]. With regard to their anatomical and behavioral features, sikas seem to compete successfully with local autochthonous deer species [38,39,46,47]. Concerning the Czech Republic, the interspecific hybridization has been documented via morphological [12,18,46] and genetic [22] analyses. Even though the spontaneous hybridization with the red deer has been known since the second half of the 19th century [12,48], it was long marginalized, belittled, or even purposely ignored [39].

Where the admixture degree is not conspicuously visible, it becomes difficult to differentiate between a standard sika/deer type and a hybrid. In such conditions, microsatellites and SNPs constitute strong DNA markers for identifying an individual or species. Regarding this fact, it should be emphasized that the main objective of our study was to quantify the genetic diversity and structure of locally adapted deer populations in western Bohemia by utilizing modern genetic techniques. For this purpose, we first collected the population data of the nuclear DNA (nDNA) markers to describe the genomic variation of the animals originating from different sites within the investigated region. Then, we searched for possible genetic admixture by using Bayesian clustering analysis. Finally, we also examined the phylogenetic analysis and constructed the NeighborNet dendrograms from the genetic distances between the sampling sites.

## 2. Materials and Methods

### 2.1. Individuals in the Study

We evaluated 123 animals from the red (*n* = 63) and sika (*n =* 60) deer populating different western and central Bohemian localities (Figure 1) within the Karlovy Vary and Central Bohemian administrative regions; in the latter case, the samples were collected in the Lány deer park. The sika deer sampling sites were as follows: the Lány deer park (*n =* 13); Žlutice (*n =* 8); the Kladská forestry enterprise (*n =* 9); Slavkovský les (*n =* 20); and Doupovské hory (*n =* 10). The red deer sampling took place in Slavkovský les (*n =* 25) and Doupovské hory (*n =* 38). The samples originated from legally hunted animals. The species were determined by the hunters, based on the morphological traits (e.g., body size, antlers, and coloration). The tissue samples were collected into plastic bags and frozen. The actual collection was carried out in accordance with the laws and ethical guidelines established and accepted in the Czech Republic.

### 2.2. DNA Extraction and Genotype Determination

The total genomic DNA was isolated from the blood or tissue samples by using the QIAamp^®^ Blood/Tissue Kit (Qiagen, Valencia, CA, USA) according to the protocol handbook. The extracted DNA was visualized on a 1% TAE agarose gel stained with ethidium bromide. The samples were then denatured at 95 °C for 7 min and kept at −20 °C until the genotyping started. We optimized a panel of 11 autosomal microsatellite markers. The alleles of the microsatellite loci *BM888*, *BM1818*, *ETH225*, *RM188*, *OarFCB5*, *RT1*, *RT13*, *T26*, *T156*, *T193*, and *T501* [49,50,51,52,53,54] were examined via multiplex polymerase chain reaction (Veriti^®^ Thermal cycler; Life Technologies, Carlsbad, CA, USA) and by applying fragment analysis (ABI PRISM 310TM Genetic Analyzer; Life Technologies, Carlsbad, CA, USA). A numerical nomenclature was employed for the allele size designation (in bp) in accordance with our internal values. We also used an internal control to calibrate the allele sizes at every run as the ISAG (International Society for Animal Genetics) comparison tests are not available to genotype nonmodel organisms in the standardization.

### 2.3. Statistical Analyses of the Genetic Data

The basic locus-specific diversity measures such as the number of alleles and their frequencies, polymorphic information content (PIC), observed heterozygosity (H_O_), and expected heterozygosity (H_E_, or gene diversity), were calculated separately for each species and locus by using PowerMarker version 3.25 [55]. As the groups being compared are species rather than true populations, they may not necessarily meet the Hardy–Weinberg expectation. The program FSTAT 2.9.4 [56] was employed to estimate the Wright’s *F*-statistics and the allelic richness (AR) by using the rarefaction method to correct differences in the population sizes. The testing of the linkage disequilibrium between all pairs of loci in each species was conducted with the same software. The frequencies of the null alleles (*F_Null_*) in each microsatellite locus were estimated via Genepop 4.2.1 [57]. The genetic distances (D_A_ [58]; or D_R_ [59]) between the studied sampling sites were computed with PowerMarker and visualized via a NeighborNet dendrogram in SplitsTree 4.13.1 [60].

We investigated the population structure and allocated the individual hybrid scores by utilizing a Bayesian clustering algorithm implemented in the software package STRUCTURE 2.3.4 [61]. The most likely number of populations in the dataset (*K*) was estimated through five independent replicates of *K* = 1–5. The model was run by utilizing a burn-in period of 5 × 10^4^ and a cycle of 15 × 10^4^ Markov chain Monte Carlo steps (MCMCs), under the standard admixed ancestry model and the correlated allele frequency model with the default parameter (λ = 1) to analyze the dataset. To categorize the animals into pure or hybrid sika and red deer, we determined whether the 95% confidence intervals for the *Q* scores overlapped at 0.01 or 0.99 in the pure sika and red deer, respectively. The research involved more than 30 observations, and the data followed an approximately normal distribution (a bell curve), meaning that we can use the *z*-distribution in the test statistics. The alpha value of *p* < 0.05 was employed to ensure statistical significance. In a two-tailed 95% confidence (or credible) interval (CI), the alpha value amounted to 0.025, and the corresponding critical value equaled 1.96. Thus, to calculate the upper and lower bounds of the confidence interval, we can take the mean ±1.96 × standard deviations from the mean. Using STRUCTURE HARVESTER v0.6.94 [62], we calculated the mean likelihood, ln *P*(*K*); the standard deviation for each value of *K*; and Δ*K*, the second-order rate of change of the likelihood with respect to *K* [63]. The results were then software processed to generate input files to be used with the CLUMPP application. CLUMPP 1.1.2 [64] then estimated the maximum rate of similarity between the *Q*-matrices during the ten replicate runs. The assignment bar plots were generated by DISTRUCT 1.1 [65]. We employed the Circos visualization software [66] and exploited a circular ideogram to facilitate global distribution of the STRUCTURE prediction power detected in each deer population, arising from a comparison of the sika-deer microsatellite data.

### 2.4. Ethical Approval

The entire sampling was performed post-mortem, involving only samples collected (independently of our research) from legally hunted red and sika deer in the Czech Republic. All applicable international, national, and/or institutional guidelines relating to the care and use of animals were followed. The procedures were carried out according to the Hunting Act (Law No. 449/2001), Decree No. 343/2015 Coll., on hunting periods for the individual game species and on detailed conditions governing hunting; for the given purpose, no specific ethical approval was required.

## 3. Results

### 3.1. Microsatellite Genetic Diversity and Population Differentiation

All *biparentally* inherited microsatellite loci analyzed in the red and sika deer were polymorphic. A total of 176 distinct alleles were observed at eleven loci over the complete dataset (*n =* 123; the *Cervus* genus). The number of alleles at the individual loci ranged from 10 (*BM1818*) to 22 (*BM888* and *T193*), with 16.00 alleles per locus on average (Table 1). The H_O_ ranged between 0.58 (*RM188*) and 0.84 (*T193*). The highest PIC was observed for the loci *T156*, *T26*, *BM888*, *RM188*, *RT13*, and *T193*; the values reached approximately 80% in the dataset of all individuals. The mean value of the major allele frequency across the loci corresponded to 0.276 in the dataset (Table 1). The number of alleles across all the samples classified into seven groups (Table 2) varied from 10.9 (Doupov red deer) to 4.27 (the Lány and Žlutice sika deer). Except for the *RT13* (0.192) and *RM188* (0.162) loci in the red deer, together with the *BM1818* (0.174) locus in the sika deer, all values of the null allele estimated frequencies were smaller than 0.1 (null allele frequencies of ≥0.2 were considered large). The mean null allele frequency values equaled 0.07 and 0.04 across the 11 loci in the red and sika deer populations, respectively (Table 3).

With regard to the average observed heterozygosity (Ho = 0.73) in the red deer, the values ranged lower than the overall average expected heterozygosity (H_E_ = 0.84); consequently, the estimated inbreeding coefficient indicated a certain level of heterozygote deficiency. The total locus-specific gene diversity in the red deer (0.70–0.91) exceeded that established in the sika deer (0.52–0.85); the observed heterozygosities and PIC exhibited similar patterns (Table 3). The Wilcoxon signed-rank test suggested that the locus-specific diversity measures estimated for the total samples reached higher values in the red deer, attaining significant levels (H_E_ and PIC at *p* < 0.01, *p* = 0.00169 and Ho at *p* < 0.05, *p* = 0.03754). The mean allelic diversity (AR with/out hybrids) ranged markedly lower in sikas (7.3/6.4) than in the red deer (12.5/11.9) across the loci (the Wilcoxon signed-rank test significant at *p* < 0.01). Populations (*Cervus nippon nippon*) with superior genetic qualities (higher Ho, H_E_, PIC, and lower *f*) were found in the subregions of Kladská, Žlutice, and Doupovské hory. The genetic variability at the Lány deer park and Slavkovský les proved to be lower (Table 2). The presence of hybrid individuals in our dataset led to a slightly overestimated genetic diversity, mainly in the red deer populations (Appendix A).

Figure 2 shows the NeighborNet dendrogram constructed from the Nei′s D_A_ distances between the seven red and sika deer sampled populations. These groups tended to cluster together according to the geography. A similar pattern appeared when we analyzed the relationships between the sampling sites, also via a NeighborNet visualization of the Reynolds’ D_R_ genetic distances in terms of short-term evolution (Appendix A).

A significant genotypic linkage disequilibrium was detected in only one pair of loci (*T501* × *RT13*) in both species; this fact then supports the hypothesis that the given loci segregate independently in each species’ genome (Appendix A).

With regard to the Wright’s *F*-statistics, the overall *F*_IS_, *F*_IT_, and *F*_ST_ values equaled 0.103, 0.273, and 0.189 (*p* < 0.05), respectively. Concerning only the non-hybrid animals, the values were as follows: *F*_IS_ = 0.098, *F*_IT_ = 0.285, and *F*_ST_ = 0.207 (*p* < 0.05). However, *F*_ST_ might prove inappropriate for comparing loci with substantially different levels of variation, could be misleading, and may yield wrong results in recently isolated populations (but still may contain some similarity due to common ancestral population). For this reason, we calculated the genetic distance, which showed much higher values (Appendix A).

### 3.2. Clustering Analysis and Interspecific Hybridization

The genetic structures of the investigated samples were evaluated by using Bayesian model-based clustering in the STRUCTURE software. The STRUCTURE analysis of the microsatellite genotypes in both species separated according to the sampling sites provided the strongest support for the grouping of the genetic variation into two clusters (*K* = 2) based on ∆*K* = 1393.14 (Figure 3a). The potential of the sika deer genetic introgression into the red deer and vice versa was revealed; the individuals were assigned to the sika deer cluster (at the membership probability of *Q* ≤ 0.01) and the red deer cluster (*Q* ≥ 0.99). Here, a “nuclear hybrid” was defined via the nuclear/microsatellite markers as an individual returning the *Q* value of 0.01 < *Q* < 0.99 between two taxa, with the confidence intervals calculated at 95%, as described in the Materials and Methods Section. Using the definitions outlined above, we found 56 pure sika deer (i.e., those with a CI overlapping 0.01), achieving an average *Q* score of 0.0025 ± 0.0015 (± SD); 54 pure red deer (CI overlapping 0.99), obtaining an average *Q* score of 0.9968 ± 0.0033; and 13 hybrid deer, where the *Q* scores ranged between 0.0193 and 0.9896. In the sika deer populations, we identified four hybrid animals at the membership probabilities of 1.93%, 2.78%, 6.78%, and 31.01%; the red deer populations then yielded nine hybrid individuals (at the membership probabilities of 98.96–94.82%) (Figure 3b).

The total hybrid score equaled 10.57% (13/123), with 14.29% (9/63) of the hybrids being among red deer–like individuals and 6.67% (4/60) among sika-like ones. In the sikas from the subregions of Kladská, Žlutice, and Lány, no hybrid animals were identified. In Doupovské hory, the total interspecific hybrid score reached 18.75% (9/48); 15.79% of the hybrids ranged among red deer–like individuals (6/38), and 30% fell within the sika-like ones (3/10). In Slavkovský les, the total hybrid score amounted to 8.89% (4/45) with 12% of red-like hybrids (3/25) and 5% of sika-like hybrids (1/20).

From the confusion matrix of the Bayesian method as calculated by STRUCTURE for each red-sika deer sampling site, we constructed a circular ideogram (Figure 4) displaying variation in the genome structures. Figure 5 indicates that the set of 123 individuals phenotypically designated as red (*n =* 63) and sika deer (*n =* 60) through genetic identification comprised 54 animals having the *Q*-values of 0.99–1.0 (“pure red deer”), nine animals exhibiting 0.75–0.99 (hybrid of the “red type”), one individual showing 0.25 ≤ *Q* ≤ 0.75 (an intermediate hybrid), three individuals with 0.01–0.25 (a hybrid of the “sika type”), and 56 animals characterized by 0–0.01 (“pure sika”). After removing all the red-sika hybrid animals, the dataset containing 110 animals was re-analyzed in STRUCTURE and clustered into two groups; the assignment test clearly isolated the red and sika populations without the admixed types.

## 4. Discussion

The hybridization between the red and sika deer was documented by using available sources including the older phenotypic and craniological [18,46] and the recent genetic [22] analyses; these references make it obvious that merely one genetic report concerning red-sika deer hybridization in the Czech Republic was completed previously. Our paper therefore most likely embodies only the second attempt at a relevant genetic analysis, performed by utilizing DNA-based methods. The research within this study exploits some of the primers developed for bovine and ovine markers, as the relevant microsatellites are often utilized in genetic analyses of the sika and other deer species, the reason being that they are conserved well in *Artiodactyla* [27,67]. Even though we used fewer neutral markers than the authors of the earlier genetic studies (11 compared to 13, 14, and 22), all of the H_E_ values detected within our sika samples (0.68) exceeded those reported formerly, considering the H_E_ = 0.48 for Poland, Russia, and Lithuania [21] besides the H_E_ = 0.47 in the case of the Czech Republic [68]. Conversely, the genetic variability values detected in the Czech sika populations ranged significantly higher than those established in other introduced sika communities across Europe (Scotland: H_E_ = 0.15 [15] and Ireland: H_E_ = 0.32 [23]). The estimated H_E_ values of the red deer in this study (0.84) were comparable to the data reported from eastern Europe (Poland, Russia, and Lithuania [21]; H_E_ = 0.79) but also higher than the values found in Ireland ([23]; H_E_ = 0.57). Smith et al. [16] also revealed that the mean allelic diversity in the red deer (10.1) exceeded that in sikas (7.1). With regard to our samples, the microsatellite diversity in the red deer (12.5) reached well above the relevant value detected in the sika deer (7.3). This pattern is not surprising and probably reflects the contrasting histories of these species in Europe: Red deer are widespread, with high levels of connectivity among the populations, whereas sikas were introduced in relatively small numbers out of limited sources [21].

Sika–red deer hybrids attracted attention already during the earliest instances of sika import into Europe. The pioneering country, the UK, accepted the first sikas in 1860 [12], and the initial written record on the hybridization appeared in 1884 [48]. The hybridization between the native red and the non-native sika deer constitutes a serious issue in the Czech Republic, even though the existence of the process has been questioned for a long time [12,19,35,39,69,70].

Concerning the presence of sikas in continental Europe, the numbers, as can be inferred from the above suggestion, were the highest in the Czech Republic. One of the largest zones of contact where sikas interbreed with red deer has formed in Doupovské hory, a broad, low-elevation mountain range that, considering the abundance of the animals, appears to have become a model for monitoring the hybridization and its effects [35]. A dedicated genetic research project executed by experts of the Czech Academy of Sciences [22] showed that the proportion of hybrids among the deer in Doupovské hory had already reached approximately 10%; such a result is lower than that presented in our study (19%). In the Pilsen region, the estimated real sika population appears to have exceeded the standardized (i.e., the utmost permissible) game stock count three- to sevenfold [39,44]. Some of the more recent reports even suggest that the authentic numbers are 43 times above the standardized stock level [69]. Such dynamic expansion is aided by the species’ significant mobility (markedly greater than in the red deer), which allows the animals to respond to food supply variations and disturbances in general [39]. Another comparatively topical analysis discussed, among other issues, is the dispersion of genetically pure red deer, indicating Krkonoše (The Giant Mountains) as one of their home locations [29].

Biedrzycka et al. [21] found 35 (15.5%) hybrid individuals in all regions (Poland, the Kaliningrad District (Russia), and Lithuania) investigated including an area where no sika deer population was traced. The extent of the gene flow between the invasive sika and the native red deer in Ireland [20,23] and Scotland [15,16,17] proved extremely variable across the different locations sampled. In Scotland, Senn and Pemberton [15] estimated the overall mixed ancestry at 6.9%, except for one site where 43% of individuals were hybrids. A corresponding study on Kintyre [16] revealed 8.9% of red-like hybrids with a recent sika ancestry and 7.1% of sika-like hybrids with a recent red ancestry. Furthermore, in Ireland, McDevitt et al. [20] found similar hybridization rates, with 9.4% and 10.6% of hybrids among red deer-like and sika-like individuals, respectively. A view of other countries lead us to assume that the introgression levels in this paper (10.6%) are comparable to those established within relevant previous studies. In Lithuania, Ražanskė et al. [24] recognized 15.9% of hybrid animals between the red and sika deer populations, based on genotyping the seven microsatellites´ loci. However, more markers allow more individuals to be assigned as hybrids (McFarlane et al. [17] compared to Senn and Pemberton [15]). McFarlane et al. [17] recently found twice as many hybrids with 44,999 SNPs (43.3% hybrids in Kintyre and 3.7% in the NW Highlands). In the same set of animals, and utilizing 22 microsatellites, Senn and Pemberton [15] had previously detected 23.2% hybrids in Kintyre and 1.9% hybrids in the NW Highlands.

Various articles characterizing the structure of the hybrid zone [8,15,16,17,23] are available; other studies then provide information on the distribution (e.g., hybrid swarm, unimodal, or bimodal) of the hybrid system [25]. In our manuscript, the Structure *Q*-scores of the hybrid cline (Figure 4) can be described as bimodal, with the deer classified into two distinct clusters, namely, red-like and sika-like (Figure 3). Our dataset comprises only one potential F1 individual. The hybrids mate with parental species’ individuals (instead of other hybrids), generating a large proportion of individuals of hybrid ancestry in the population; such animals are difficult to distinguish, either genetically or phenotypically, from the parental species. It is important to consider that some species can also form a hybrid zone, and this cline can be relatively stable [1,71].

Surprisingly, in contrast to the literature, we detected low genetic differentiation between the species (*F*_ST_ = 0.21, *p* < 0.05). In this context, a lower *F*_ST_ value denotes similarity between the allele frequencies within each red and sika deer population; contrary to that, the *F_ST_* between the red deer and the sika deer in Scotland was ~0.5 [15,17]. In Kadyny forest, Biedrzycka et al. [21] established a somewhat smaller genetic differentiation between sikas and other populations, that of the red deer in particular (the *F*_ST_ values ranged from 0.33 to 0.37). Regrettably, some authors [16,20,23,24] did not indicate any *F*_ST_ values. Within our paper, the partial values were low, but the global *F*_ST_ significantly differed from zero, suggesting the presence of a population structure. Comparing the *F*-statistics inside a species can also be a demanding task, especially when separate parts of the distribution differ in diversity. The populations having generally high diversities showed lower levels of divergence as measured by *F*_ST_, while those with low diversities exhibited higher *F*_ST_ values. High diversity populations are expected to have low divergences, as variation within a population may indicate outcrossing with other populations, and high variation also makes these populations more prone to sharing variability with others at random.

The hybridization and changes in behavior embody the most important problems between the two species living in sympatry. The timing and synchronization of the rut were analyzed by Macháček et al. [72], who covered, above all, Doupovské hory; the investigation proved that both species experienced a shift in the rutting period (the rutting season starts later in the red and earlier in the sika deer), and this effect probably embodies another manifestation of continuous hybridization. Likewise, the peak rut in sika hinds became a focus for Ježek et al. [37], whose research to a large extent overlapped territorially with that outlined in the above paper; the proposed closer view on the individual areas, namely, the western Bohemian microregions, then revealed that the discussed stage of rut set first in actual Doupovské hory, during the initial ten days of October. Such processes associated with the shift may also have influenced the number of hybrids determined through our investigation of this concrete geographical subsector. Furthermore, exclusively in relation to the same location, the results also indicate a major increase in the weight of the hinds. Weight analyses targeting sikas and the red deer were delivered by various authors including J. Křivánek, a specialist on the deer populations of Doupovské hory; a dedicated article [35] suggests that in the microregion, the greatest weight variations in both sexes occurred after 1989, with the dressed weight gradually rising in the sika and falling in the red deer. This most likely embodies another element to manifest ongoing hybridization. The coefficient of expected production, which is to express the number of fawns per doe a year, equaled 0.945 in the sika deer across the Czech Republic [37].

With regard to the hybrids, interesting details were recently revealed by several research teams [73,74]. Wyman et al. [73] showed that neither red deer nor sika hinds preferred pure-bred calls over hybrid ones, the reason being that the females are unable to easily distinguish, in this respect, between a pure-bred and a hybrid stag, thus increasing the potential for further introgression. Li et al. [74], by extension, found out that the rumen microbiota of the red-sika hybrids differed from that of their parents, suggesting a significant impact of host genetics on the rumen microbiome; such an impact may result from vertical transmission.

The advantage of identifying species at the molecular level rests in that DNA is inalterable and detectable in every cell. Molecular marker-based identification is possible even with small amounts of biological tissue or germplasm (including ova and semen), which do not show an evident phenotype. Genomic data analysis nevertheless also brings difficulties, for instance, more markers increase the probability that more individuals will be assigned as hybrids. McFarlane et al. [17] recently identified nearly twice as many hybrids than previously detected in the Scottish red deer x sika hybrid zone (McFarlane et al. [17] compared to Senn and Pemberton [15]); the result was obtained by using ~45 k markers compared to 22 diagnostic microsatellites. Discussing microsatellite markers, we need to note that they are informative in detecting first-generation or second-generation hybridization events, but cannot reveal extensive backcrossing over several generations between parental species [25,26]. An illustrative example was provided by Vähä and Primmer [75], who ran a number of simulations to evaluate how many markers are required for assigning hybrids and parental species. With an *F*_ST_ of ~0.21 and 12 polymorphic loci, the authors estimated a performance (the assignment of hybrids as hybrids and parentals as parentals) of ~55%. McFarlane et al. [17] noticed that not more than three generations of backcrossing are reliably detectable with 10 markers. Given that in our study ~120 years may have elapsed since a secondary contact, there is a potential for substantial backcrossing, which would be difficult to find with the markers. To improve the accuracy, we intend to apply a higher number of nDNA markers to reclassify the individuals examined herein. Ideally, a further intensive nationwide sampling-based study to assess the real impact of crossbreeding on the sika and red deer genepools appears to be a potentially fruitful plan; nevertheless, the possibility of hybridization with other deer species (maral, wapiti) or sika deer subspecies has to also be verified. In the given context, the research would also benefit from comprehensive attempts to identify asymmetric introgression of the maternal species contribution (mitochondrial markers) into a paternal species background.

With regard to the visible phenotypic traits, first-generation crossbreeds exhibit a prevalence of sika features, although the resulting animals are somewhat larger than standard sikas (but, at the same time, less sizeable than the red deer); second-generation and subsequent individuals (backcrosses), however, essentially cannot be morphologically distinguished from the source species in cases of repeated sika-red deer hybridization [12,18,19]. Finally, it is questionable whether detecting such advanced backcrosses involves a conservation management value. More practically, a mixed strategy of preserving those individuals that are genetically parental species (i.e., narrow *Q* score CIs that overlap with 0 or 1) and score highly on emblematic red deer phenotypes (e.g., summer coat without spots, long pointed ears, large antlers) could reduce the threat to red deer from sika introgression [17].

Sikas and their positives and negatives have been analyzed multiple times thus far. On one hand, the sika deer is considered a convenient substitute for the red deer in areas where the native species has disappeared due to landscape disturbances; many gamekeepers have begun to openly prefer sikas, adjusting the hunting grounds and regulations to suit the animals, whose benefits include delicious meat, a good trophy status, high intelligence, modest raising and keeping requirements, superior resilience, and general adaptability. On the other hand, sikas, similarly to other introduced cloven-hoofed game, interfere with the indigenous species and, through crossbreeding, endanger the gene pool of the red deer.

## 5. Conclusions

In the Japanese sika deer, the high potential to compete with the autochthonous species and the readiness to hybridize with the native red deer have posed a real threat to the original European red deer populations for over a century. In the Czech Republic, the hybridization between the native red deer and the non-native sika deer constitutes a problem largely ignored to date, and its very existence has been questioned for a long time. Using microsatellite DNA markers, we investigated the genetic structure and hybridization events in the sika and red deer populations from western Bohemia, where sikas have proliferated to establish self-sustaining communities. The fixation index value indicates small but important genetic differentiation between the two main genetically distinct species. Regarding the content and methodology of the research, we found 13 nuclear sika-red deer hybrid forms at two sites. We confirmed the crossbreeding in Doupovské hory and Slavkovský les; the highest interspecific hybridization value was found in the former of these subareas (19%). In the given context, we detected more red-like hybrids than sika-like ones. Within the article, an individual returning a *Q* value of 0.01 < *Q* < 0.99 was identified as a ‘nuclear hybrid’, based on the microsatellite markers. Animals outside these boundaries were defined as ‘pure’, although they may still have contained introgressed alleles beyond the detection limit of the markers. Regrettably, comparing the incidence of hybrids in the region and considering the number of samples and genetic markers used, it is well possible that many later generation backcrosses remained undetected.

## Figures and Tables

**Figure 1 animals-11-01701-f001:**
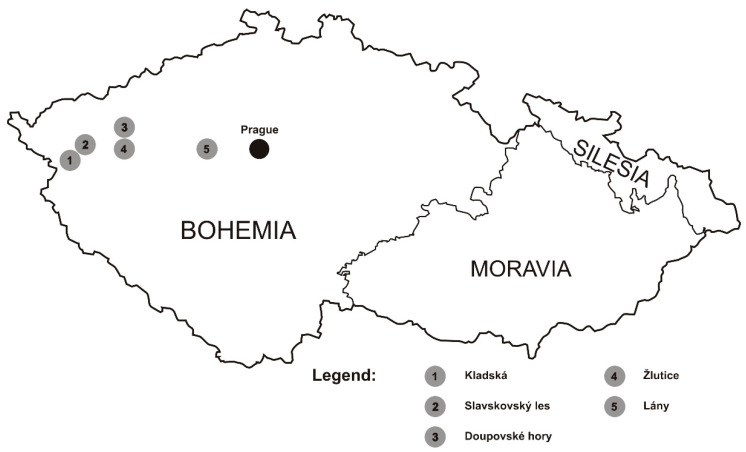
A map of the Czech Republic showing the sampling locations in the red and sika deer.

**Figure 2 animals-11-01701-f002:**
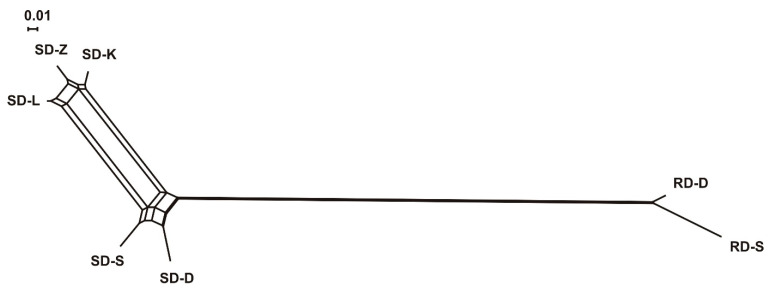
The NeighborNet dendrogram constructed from the Nei´s DA distances between the seven red and sika deer sampling populations (*n =* 123). Population codes: RD-D (Red deer Doupovské hory), RD-S (Red deer Slavkovský les), SD-D (Sika deer Doupovské hory), SD-S (Sika deer Slavkovský les), SD-K (Sika deer Kladská), SD-L (Sika deer Lány), and SD-Z (Sika deer Žlutice).

**Figure 3 animals-11-01701-f003:**
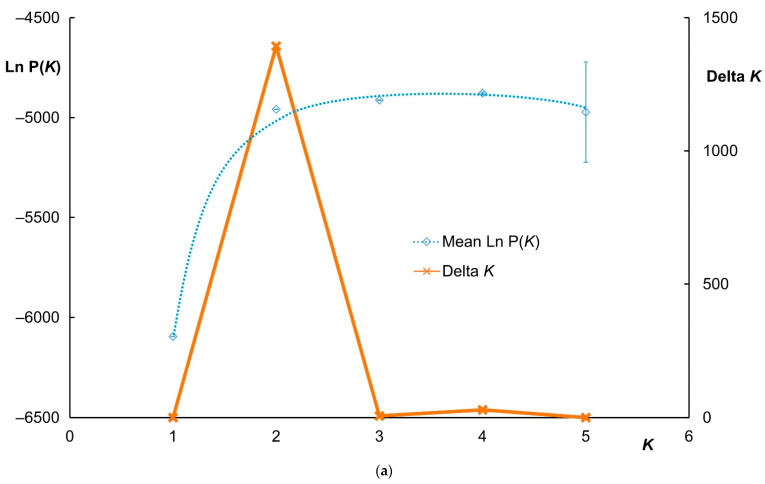
The Bayesian model-based clustering rendered with the STRUCTURE software: (**a**) the evolution of the mean ln of likelihood (ln P(*K*)) according to *K*, based on the five runs of the 50,000 burn–ins and 150,000 MCMCs (standard deviations indicated); (**b**) STRUCTURE clustering results at *K* = 2 (the admixture model and 11-locus dataset). Each vertical line represents one individual. The thin black lines separate individuals from different sampling sites (groups). Population codes: RD-D (Red deer Doupovské hory), RD-S (Red deer Slavkovský les), SD-D (Sika deer Doupovské hory), SD-S (Sika deer Slavkovský les), SD-K (Sika deer Kladská), SD-L (Sika deer Lány), and SD-Z (Sika deer Žlutice).

**Figure 4 animals-11-01701-f004:**
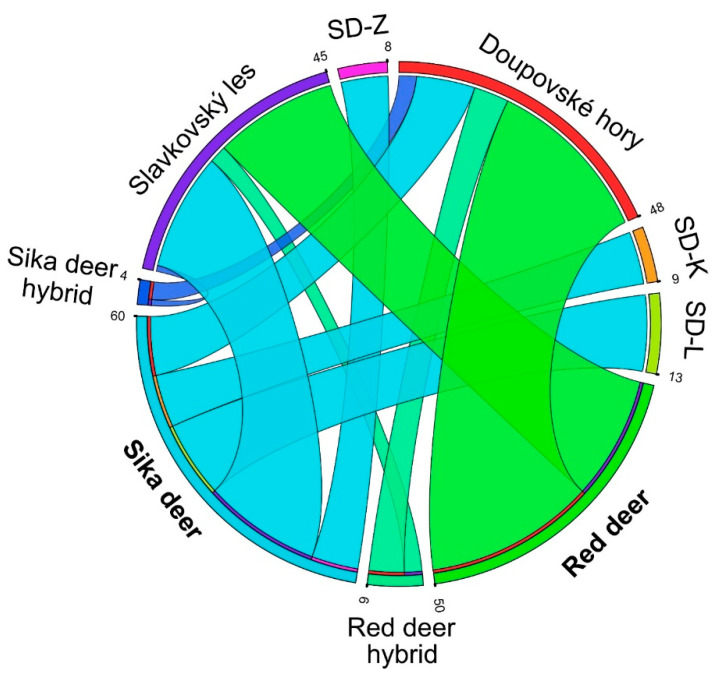
The Circos-like plot displaying the interspecies variation based on the nDNA data. Population codes: SD-K (Sika deer Kladská), SD-L (Sika deer Lány), and SD-Z (Sika deer Žlutice).

**Figure 5 animals-11-01701-f005:**
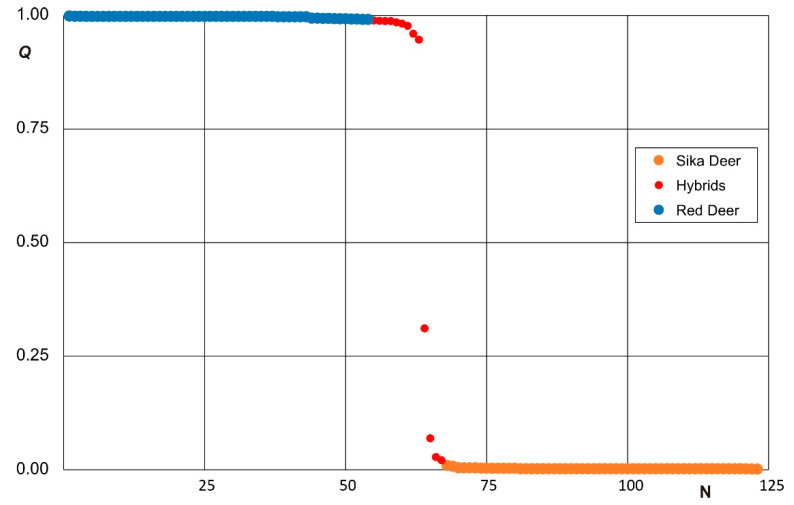
The estimated proportion of interspecific hybridization ancestry (*Q*) between the sika and red deer (*n =* 123), and the threshold values of 0.01 < *Q* < 0.99 used to detect the hybrid individuals in the western parts of the Czech Republic. We detected more red-like hybrids (*n =* 9) exhibiting a recent sika ancestry (0.50 < *Q* < 0.99) than sika-like ones (*n =* 4) having a recent red deer ancestry (0.01 < *Q* < 0.50).

**Table 1 animals-11-01701-t001:** The summary statistics of the microsatellite loci across all samples (*n =* 123).

Marker	MAF	Genotype No.	No. of Observations	Allele No.	Availability	H_E_	H_O_	PIC	*F*
*OarFCB5*	0.4024	34	123	13	1.0000	0.7781	0.6992	0.7561	0.1055
*T156*	0.2125	55	120	17	0.9756	0.8996	0.8167	0.8920	0.0964
*T26*	0.1595	48	116	16	0.9431	0.9052	0.7586	0.8975	0.1661
*BM888*	0.2500	50	120	22	0.9756	0.8739	0.6917	0.8628	0.2125
*RM188*	0.2479	31	117	17	0.9512	0.8650	0.5812	0.8521	0.3319
*RT1*	0.2764	31	123	13	1.0000	0.8211	0.6423	0.7998	0.2217
*T501*	0.3696	34	115	14	0.9350	0.8050	0.6000	0.7868	0.2587
*RT13*	0.2059	43	119	17	0.9675	0.8737	0.5966	0.8612	0.3209
*T193*	0.2292	49	120	22	0.9756	0.9034	0.8417	0.8971	0.0725
*BM1818*	0.3824	30	119	10	0.9675	0.7943	0.6471	0.7745	0.1894
*ETH225*	0.2967	39	123	15	1.0000	0.8162	0.6585	0.7941	0.1971
Mean/Overall	0.2757	40.3636	119.5455	16	0.9719	0.8487	0.6849	0.8340	0.1971

MAF = major allele frequency; H_E_ = expected heterozygosity; H_O_ = observed heterozygosity; PIC = polymorphic information content; *F* = within-sampling site inbreeding coefficient.

**Table 2 animals-11-01701-t002:** The population summary statistics across the red and sika deer sampling sites.

Population	Code	MAF	Genotype No.	*n*	No. of Observations	Allele No.	Availability	H_E_	H_O_	PIC	*F*
Red deer Doupovské hory	RD-D	0.2897	22.2727	38	36.8182	10.9091	0.9689	0.8276	0.7454	0.8096	0.1129
Red deer Slavkovský les	RD-S	0.2866	16.9091	25	24.1818	10.3636	0.9673	0.8242	0.7078	0.8057	0.1619
Sika deer Doupovské hory	SD-D	0.4742	6.3636	10	9.7273	5.7273	0.9727	0.6621	0.6364	0.6240	0.0930
Sika deer Slavkovský les	SD-S	0.5323	7.3636	20	19.2727	5.2727	0.9636	0.6088	0.5564	0.5612	0.1126
Sika deer Kladská	SD-K	0.4380	5.1818	9	8.8182	4.3636	0.9798	0.6806	0.8398	0.6254	−0.1766
Sika deer Lány	SD-L	0.5584	5.7273	13	12.7273	4.2727	0.9790	0.5850	0.5757	0.5345	0.0574
Sika deer Žlutice	SD-Z	0.4659	4.9091	8	8.0000	4.2727	1.0000	0.6584	0.7159	0.6052	−0.0208

MAF = major allele frequency; H_E_ = expected heterozygosity; H_O_ = observed heterozygosity; PIC = polymorphic information content; *F* = within-sampling-site inbreeding coefficient.

**Table 3 animals-11-01701-t003:** The locus-specific diversity measures estimated for the total sample sizes of the red (*n =* 63) and sika deer (*n =* 60).

Marker	AR		H_E_		H_O_		PIC		*F_Null_*	
	Red Deer	Sika Deer	Red Deer	Sika Deer	Red Deer	Sika Deer	Red Deer	Sika Deer	Red Deer	Sika Deer
*OarFCB5*	11.690	6.982	0.8644	0.5986	0.8254	0.5667	0.8416	0.5658	0.0300	0.0357
*T156*	12.484	9.931	0.8846	0.8352	0.8065	0.8276	0.8650	0.8107	0.0493	0.0000
*T26*	12.789	10.724	0.8921	0.8505	0.8103	0.7069	0.8732	0.8240	0.0412	0.0832
*BM888*	16.653	6.991	0.8417	0.6915	0.7333	0.6500	0.8182	0.6514	0.0767	0.0120
*RM188*	9.675	8.000	0.7873	0.6731	0.5079	0.6667	0.7508	0.6397	0.1624	0.0179
*RT1*	11.390	5.899	0.6989	0.6423	0.6984	0.5833	0.6668	0.5980	0.0266	0.0474
*T501*	10.000	9.492	0.8111	0.5363	0.6607	0.5424	0.7859	0.5038	0.1126	0.0000
*RT13*	15.721	6.931	0.8824	0.7652	0.5410	0.6552	0.8637	0.7242	0.1917	0.0630
*T193*	14.771	6.900	0.9090	0.7182	0.9000	0.7833	0.8929	0.6767	0.0000	0.0000
*BM1818*	8.000	4.000	0.8291	0.6628	0.7742	0.5088	0.8037	0.5933	0.0086	0.1740
*ETH225*	13.810	4.791	0.8555	0.5167	0.7778	0.5333	0.8337	0.4120	0.0626	0.0000
Mean/Overall	12.453	7.331	0.8415	0.6809	0.7305	0.6386	0.8178	0.6363	0.0692	0.0394

AR = allelic richness; H_E_ = expected heterozygosity; H_O_ = observed heterozygosity; PIC = polymorphic information content; *F_Null_* = estimated frequencies of null alleles for each microsatellite locus.

## Data Availability

The microsatellite data presented in this study are available in the Appendix A.

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
