# Peer review of "A Microsatellite Genotyping-Based Genetic Study of Interspecific Hybridization between the Red and Sika Deer in the Western Czech Republic"

_animals, 2021, doi:10.3390/ani11061701_

Round 1

Reviewer 1 Report

The study of Štohlová-Putnová and co-authors tries to expand our knowledge on the effect that invasive species can have on the native species, by exploring the genomic introgression by cross-species hybridization occurred in the wild between the red and sika deer in regions of the Czech Republic.

Minor comments:

Please add a map with sampling localities and indicate the locality names that you use in text and in figure 4. Please keep in mind that the majority of the readers are not familiar with the geography of Bohemia.

Line 25: please rephrase “next-generation hybrids” to “inter-species hybrids”

Line 26: although technically correct, by “nuclear DNA markers” the majority of the readers would assume nuclear genes, so, in order to avoid confusion, please rephrase to “microsatellite markers”

Lines 192: typo at “Ethical approva”

Author Response

Dear Reviewer 1,

We thank you for your time and valuable effort enabling us to improve the manuscript, and we hope that the comments have been duly addressed and the relevant problems clarified.

We look forward to your response.

Sincerely,

the Authors

Reviewer 2 Report

Reading this article, one gets the impression that sika deer is the greatest danger for biodiversity ever recorded. However, since it has been introduced to the Czech Republic at the end of the 19th century, the extent of hybridisation does not seem to be that large – based on the authors’ own results. It is important to consider that some species can also form a hybrid zone and this cline can be relatively stable (see, for instance, Beysard & Heckel 2014).

Specific comments:

L55: Would you eliminate all hybrid individuals, including backcrosses? Or only those who display are genetically more related to sika than to red deer? What would be the cut-off value for classifying an individual as a hybrid? With what markers? This is a contentious topic and I suggest being careful with such blank statements.

L56-57: This is simply not true. Genetic tools have been commonly implemented to assess hybridization for many years (e.g. Aboim et al. 2010; Boyer et al. 2008; Carneiro et al. 2016; Leache & Cole 2007).

L57: Only in the discussion section it is finally mentioned that microsatellites may not be informative enough, because previous studies have shown that SNPs are much better at detecting hybridization. This of course raises the question: why did you use microsatellites in the first place if you knew that your results will likely be inconclusive?

L65-67: This is true for both the native and introduced species.

L69-70: Add a Latin name.

L103: What are these related benefits?

L104: This is a repetition of L75.

L108-110: Is this damage greater than what is caused by red deer? And if so, how has this been quantified? Please discuss.

L112-113: This sounds like your study is a mere reiteration of what has already been done before. What is novel here?

L117: The term “admixtured hybrid” is not a proper expression – both grammatically and scientifically.

L131-139: Please include a figure depicting the sampling locations on a map.

L132: More details on how were the species determined is needed. I assume it was based on morphological traits, but what specifically?

L137: Again, more details on the sampling needs to be provided.

L144-145: This is an unusual step. Please explain why the samples had to be denatured.

L145-146: In general, it would be recommended to use approximately 20 microsatellite markers for such analyses. Why did you implement only 11?

L159-160: Indeed, and this is an important point to keep in mind also in your discussion.

L168-170: I strongly encourage you to go back to your data and conduct additional analyses with both other Bayesian clustering methods and other techniques. Running the data through Structure is the bare minimum and there surely is much more information that you could get out of your samples.

L191: Figure 2 should be below this line, to keep it close to where in the text it was first mentioned.

L192-195: This statement is again not detailed enough. Did you shoot the deer? Did you kill the animals yourselves or with a collaboration with an expert?

More importantly, I would like to ask the authors: why did you not implement non-lethal methods in your work?! Genetic analysis can be easily done using non-invasively collected samples, such as faeces. This is not a new invention, but an approach that has been implemented for at least 30 years by now (Alasaad et al. 2011; Baumgardt et al. 2013; de Oliveira et al. 2020; Hoss et al. 1992; Idaghdour et al. 2003; Sastre et al. 2009). The estimation of morphological traits could have been done through camera traps (Gray 2018; Tape & Gustine 2014). Yes, the authors got a permit to shoot the deer, but that does not mean that this was the best approach. Since the authors expressed concern about red deer populations, then why did they choose to use a lethal method? Such a study design goes not only against logic but against research ethics as well.

L320: Please discuss the potential reasons for this observation in more detail.

L381-385: Then why didn’t you the SNPs in the first place?

L427: This is not entirely true. DNA is very sensitive to high temperatures.

L445-451: I suggest that you do this first, compare it to the results you have now, and then try to publish it. Such a study would have a much higher informational value.

L475-476: Based on your results, the statement that sikas are currently “a major problem” is not convincing.

General comments:

Please check your English. Some expressions are rather awkward, for example, “density was discontinued” on L71, “to have set a model for monitoring” on L93, “saw the sikas” on L335, “arrives first in the actual Doupovske hory” on L410.  

Figure 4 might look nice, but I don’t see the informational value.

References

Aboim MA, Mavarez J, Bernatchez L, Coelho MM (2010) Introgressive hybridization between two Iberian endemic cyprinid fish: a comparison between two independent hybrid zones. Journal of Evolutionary Biology 23, 817-828.

Alasaad S, Soriguer RC, Jowers MJ, Marchal JA, Romero I, Sanchez A (2011) Applicability of mitochondrial DNA for the identification of Arvicolid species from faecal samples: a case study from the threatened Cabrera's vole. Molecular Ecology Resources 11, 409-414.

Baumgardt JA, Goldberg CS, Reese KP, Connelly JW, Musil DD, Garton EO, Waits LP (2013) A method for estimating population sex ratio for sage-grouse using noninvasive genetic samples. Molecular Ecology Resources 13, 393-402.

Beysard M, Heckel G (2014) Structure and dynamics of hybrid zones at different stages of speciation in the common vole (Microtus arvalis). Molecular Ecology 23, 673-687.

Boyer MC, Muhlfeld CC, Allendorf FW (2008) Rainbow trout (Oncorhynchus mykiss) invasion and the spread of hybridization with native westslope cutthroat trout (Oncorhynchus clarkii lewisi). Canadian Journal of Fisheries and Aquatic Sciences 65, 658-669.

Carneiro J, Rodrigues LFD, Schneider H, Sampaio I (2016) Molecular data highlight hybridization in squirrel monkeys (Saimiri, Cebidae). Genetics and Molecular Biology 39, 539-546.

de Oliveira ML, Peres PHD, Gatti A, Morales-Donoso JA, Mangini PR, Duarte JMB (2020) Faecal DNA and camera traps detect an evolutionarily significant unit of the Amazonian brocket deer in the Brazilian Atlantic Forest. European Journal of Wildlife Research 66.

Gray TNE (2018) Monitoring tropical forest ungulates using camera-trap data. Journal of Zoology 305, 173-179.

Hoss M, Kohn M, Paabo S, Knauer F, Schroder W (1992) Excrement analysis by PCR. Nature 359, 199-199.

Idaghdour Y, Broderick D, Korrida A (2003) Faeces as a source of DNA for molecular studies in a threatened population of great bustards. Conservation Genetics 4, 789-792.

Leache AD, Cole CJ (2007) Hybridization between multiple fence lizard lineages in an ecotone: locally discordant variation in mitochondrial DNA, chromosomes, and morphology. Molecular Ecology 16, 1035-1054.

Sastre N, Francino O, Lampreave G, Bologov VV, Lopez-Martin JM, Sanchez A, Ramirez O (2009) Sex identification of wolf (Canis lupus) using non-invasive samples. Conservation Genetics 10, 555-558.

Tape KD, Gustine DD (2014) Capturing migration phenology of terrestrial wildlife using camera traps. Bioscience 64, 117-124.

Author Response

Dear Reviewer2,

We thank you for your time and valuable effort enabling us to improve the manuscript, and we hope that the comments have been duly addressed and the relevant problems clarified.

We look forward to your response.

Sincerely,

the Authors

Reviewer 3 Report

The manuscript “A microsatellite genotyping-based genetic study of interspecific hybridization between the Red and Sika deer in western Bohemia” by Štohlová-Putnová et al is one of the higher quality papers that I have reviewed in the last several months. It is well formatted and its English is of superior quality. I find the methodology correct, and I had few issues with the conclusions to which the authors came. They analyzed a realistically large population of Sika and Red deer (>100) with a practical number of polymorphic markers (11) for genotype studies using accepted methods. Additionally, the authors give sufficient historical detail in the Introduction to make the purpose of their study have ecological bearing. They have a goal of being able to identify Red/Sika deer hybrids using molecular techniques, leading to better environmental practices to reduce the invasive Sika population and increase the variability and genetic health of the Red deer.  The authors state this very clearly as: “the relevant objective of our study is to quantify the genetic diversity and structure of locally adapted deer populations in western Bohemia, utilizing state-of-the-art genetic techniques.”

However the paper is not perfect, and needs to have a few things fixed. I recommend minor revisions.

The Title: I am unsure why the title refers to “Western Bohemia” instead of  “The Western Czech Republic”.  Unless I am very mistaken, Bohemia seems to be an outmoded historical term and would just be confusing to readers who are unfamiliar with older, Eastern European geography. I recommend a change in the title to the Western Czech Republic.

Figure?: It is pretty standard in a geographical studies such as this one to include a map which shows both a close-up of the area of interest and a wider, more panoramic view to give the reader perspective. Again, not all readers are familiar with Eastern European geography. I would recommend inclusion of a new map showing the geographical locations discussed throughout this paper.

Abstract, Line 25: You state: “…there is the possibility of quick identification.”  Unless I am misinterpreting your meaning, stating there is a “possibility” of success seems strange since you succeeded.

Line 121: Some would argue whether microsatellite analysis is “state of the art.” I employed microsatellite analysis 25 years ago. I would suggest that using NGS phylo-analysis in which 50 polymorphic markers are examined simultaneously is actually state of the art, but that’s just quibbling.

Materials & Methods, Line 132: Please give some more detail on how deer were selected and how blood or tissue samples were actually collected.

Author Response

Dear Reviewer 3,

We thank you for your time and valuable effort enabling us to improve the manuscript, and we hope that the comments have been duly addressed and the relevant problems clarified.

We look forward to your response.

Sincerely,

the Authors

Reviewer 4 Report

            This manuscript presents a comprehensive assessment of hybridization between red and sika deer in Czech Republic. Hybrids between both species are difficult to recognize based on morphology, so the authors use a series of microsatellites for detection. The molecular data also allows to characterize population structure in both species across the study area. One of the main results is the detection of hybrids in both species with relatively a similar hybrid score than in other parts of Europe. The study does a good job describing the need for this data and providing the background information. In the same way, the methodological description is clear and complete, and results appropriately described. But, the Discussion can be improved by focusing more on the implications of the presented results, further discussing some of them (ie. Low level of admixture in the detected hybrids) and moving several sections to the Introduction. In addition, there are many small comments that need to be addressed throughout the text. My assessment is “Minor revisions”.

General comments

  • The abstract would benefit of a conclusion sentence at the end.
  • The big picture of the study could be extended a bit more by presenting the implications of hybridization for conserving native species. Perhaps, presenting a couple of examples beyond deer. Currently, there are only 3 lines of broad introduction before jumping into the study system. A couple of more general sentences could better motivate the relevance of the study.
  • The general writing is good, however some aspects could be improved. In particular, (1) some paragraphs are too long including different main ideas, and (2) the beginning of some phrases appear to have unnecessary introductory sentences, which are more appropriate for a conversational situation. Below there are specific suggestions to address some sections of the text with these issues.
  • The description of the situation in Czech Republic is very well documented. In Line 92 there is a brief mention to the situation in the UK. To motivate the broader interest of the paper, I would recommend include references to other areas where sika deer introduction has happened. In particular, referring to the cases where hybridization has been documented. This information is currently present in Lines 366-371. I suggest moving that section to the Introduction.
  • It would be useful to include a map with the sampling sites. That would give a better geographic context for the readers not familiar with the geography of the Czech Republic.
  • In Line 144, the authors say “The samples were then denatured at 95 C for 7 min and kept at -20 °C until the genotyping started.” I am not familiar with denaturated storage of DNA samples before PCR. Could the authors add an explanatory sentence and a reference to for this procedure?
  • 1 and S1: Did the authors only use one representative per population on the NeighborNet dendrogram? If yes, they should explicitly mention that on the Methods.
  • NeighborNet dendrogram: It would be interesting to see the same NeighborNet dendrogram using all the individuals. This could be used as an independent way to provide evidence of hybrid detection. This is just an optional suggestion.
  • Line 340-355: This section would be better placed in the Introduction. Summarize it here and leave only the central idea for discussing your results.
  • Line 402-418: These is interesting natural history information, but it is not clear the relationship with the presented results. I would suggest to focus more in the time differences in the rut of both species. That could be used as an explanation to the absence of even more hybrids. Moreover, it could be discussed what would be the consequences of an increase in the temperatures due to global warming.
  • Line 429: “Genomic data analysis nevertheless has its drawbacks, too: For instance, more markers increase the probability that more individuals will be assigned as hybrids.” I do not understand why this could be considered a drawback. Indeed, I think it is the opposite. Deeper sequencing provides a more complete picture of the situation. Reconsider and rewrite the statement.
  • Line 451: “…not only nuclear red deer-sika hybrids but also mitochondrial ones.” There are no mitochondrial hybrids, instead it is possible to detect asymmetric introgression of the maternal species contribution (mitochondrial markers) into a paternal species background. Please, re-write accordingly.
  • Line 452-457: Move to the Introduction.
  • Line 463-470: It is an interesting paragraph, relevant for the management of both species. However, it needs a couple of introductory sentences to link it with the rest of the discussion. Please, add them.
  • Line 477: “…hybridization between the native red deer and the non-native sika deer constitutes a major problem” Considering the similar % of hybridization with other areas and the fact that they tend to be around 10%, I would not argue that it is a “major problem”. Instead, I would describe it as a “neglected problem” or “largely ignored problem”. I would focus, as appears on the discussion, on the potential implications of this and the fact that, until this paper, it has not been widely studied. My proposed focus fits with the second part of the sentence (which I think should be maintained): “…even though its actual existence has been questioned for a long time.” I think a “major problem” would be justified if hybrids appeared to be more common. Note, this is not a strong suggestion. I leave up to the authors to accept it or not.
  • It is very important to further discuss the sharpness of the hybrid cline (Fig. 3) and the low % mixed genetic background (Fig. 2). Considering the fact that this implies older hybridization events. What are the implications regarding the overall frequency of hybridization events?

Minor comments:

  • Line 24: I am not familiar with the term “next-generation hybrids”. Could you further explain, or change for a more common concept? Just “hybrids”, maybe.
  • Line 44: Replace “anthropogenic hybridization” by “anthropogenic-induced hybridization”
  • Line 78: Could you provide a one sentence explanation of that population decline towards the present?
  • Line 112: Replace “…hybridization was documented…” by “..hybridization has been documented…”
  • Line 119: Replace “relevant” by “main”
  • Line 121-122: I would recommend to replace “Procedurally, the research can be characterized as follows: At the initial stage, we…” by “In order to do that, we first …”. This is not a strong suggestion. I leave up to the authors to accept it or not.
  • Line 124: Replace “Subsequently” by “Then”. This is not a strong suggestion. I leave up to the authors to accept it or not.
  • Line 124-125: Replace” we attempted to detect possible degrees of genetic admixture…” by “we search for potential genetic admixture…”
  • Line 126: Delete “In the given context,”
  • Line 126: Replace “…sought to construct…” by “construct”
  • Line 131: Replace “experiment” by “study”.
  • Line 138: Replace “All the experiments” by “The specimen collection”
  • Line 144: Remove “…yielding the relevant quality and quantity.” It is unnecessary to state.
  • Line 218: Replace “…higher in…” by “…higher values in…”
  • Line 222: What does it mean “best genetic qualities”? Please, define.
  • Line 224: Replace “somewhat worse” by “lower”, if that was the intention of the sentence.
  • Fig. 4 is cited before Fig. 3. Then, switch the figure numbers.
  • Line 315: Delete “molecules”
  • Line 418: Starting on the following sentence (“As regards the hybrids, interesting details…”), the rest of the paragraph could be moved into a separate paragraph.

Author Response

Dear Reviewer 4,

We thank you for your time and valuable effort enabling us to improve the manuscript, and we hope that the comments have been duly addressed and the relevant problems clarified.

We look forward to your response.

Sincerely,

the Authors

Round 2

Reviewer 1 Report

The manuscript was improved after the the last round of revision, and the authors replied satisfactory to all my commnets, therefore, I have no additional comments.

Author Response

Dear Reviewer,

We thank you for your time and valuable effort.

Sincerely,
the Authors

Reviewer 2 Report

Dear authors, thank you for your response and revisions.

It is good to hear that the animals were not killed for the purposes of this research. You also addressed most of my comments satisfactorily.

However, I still have remaining concerns when it comes to your analysis. In the previous round of review, I stated:

L168-170: I strongly encourage you to go back to your data and conduct additional analyses with both other Bayesian clustering methods and other techniques. Running the data through Structure is the bare minimum and there surely is much more information that you could get out of your samples.

To which your response was:

Regrettably, we are unable to incorporate other techniques at present, due to the limited response time. As regards Structure, the program was employed as the sole tool also by other authors that focused on the hybridization and needed to perform relevant analyses (Senn and Pemberton 2009, Smith et al. 2014, Smith et al. 2018, McFarlane et al. 2020).

Senn, H.V.; Pemberton J.M. Variable extent of hybridization between invasive sika (Cervus nippon) and native red deer (C. elaphus) in a small geographical area. Mol. Ecol. 2009, 18, 862–876. https://doi.org/10.1111/j.1365-294X.2008.04051.x.

Smith, S.L.; Senn, H.V.; Pérez‐Espona, S.; et al. Introgression of exotic Cervus (nippon and canadensis) into red deer (Cervus elaphus) populations in Scotland and the English Lake District. Ecol. Evol. 2018, 8, 2122–2134. https://dx.doi.org/10.1002%2Fece3.3767.

McFarlane, S.E.; Hunter, D.C.; Senn, H.V.; et al. Increased genetic marker density reveals high levels of admixture between red deer and introduced Japanese sika in Kintyre, Scotland. Evol. Appl. 2020, 13, 432-441. https://doi.org/10.1111/eva.12880.

Smith, S.L.; Carden, R.F.; Coad, B.; et al. A survey of the hybridisation status of Cervus deer species on the island of Ireland. Conserv. Genet. 2014, 15, 823-835. https://doi.org/10.1007/s10592-014-0582-3.

I have checked all of the papers that you listed and one big difference between their methodology and yours is that all the other studies used also another marker additionally to microsatellites – either mitochondrial marker or SNPs. That is why they can get away with using just STRUCTURE because they include other types of analysis that could be compared to the results from microsatellite markers.

Additional analyses of your current data and/or conducting additional laboratory work to sequence your samples with an mtDNA marker would strongly improve your study. It would also enable you to draw more convincing conclusions.

It is regrettable that you consider time constraints as a barrier to conduct additional analyses and get the full potential from your data.

Author Response

(The authors gave the same response as above.)
